# Effectiveness of Swallowing Care on Safe Oral Intake Using Ultrasound-Based Observation of Residues in the Epiglottis Valley: A Pragmatic, Quasi-Experimental Study

**DOI:** 10.3390/healthcare8010050

**Published:** 2020-02-27

**Authors:** Mikako Yoshida, Yuka Miura, Shingo Okada, Masako Yamada, Hitoshi Kagaya, Eiichi Saitoh, Yayoi Kamakura, Yohei Okawa, Yutaka Matsuyama, Hiromi Sanada

**Affiliations:** 1Department of Women’s Health Nursing & Midwifery, Tohoku University Graduate School of Medicine, 2-1 Seiryo-machi, Aoba-ku, Sendai, Miyagi 9808575, Japan; mokka-tky@umin.ac.jp; 2Department of Imaging Nursing Science, Graduate School of Medicine, The University of Tokyo, 7-3-1 Hongo, Bunkyo-ku, Tokyo 1130033, Japan; yukam-tky@umin.ac.jp; 3Kitamihara Clinic, 350-18 Ishikawa-cho, Hakodate, Hokkaido 0410801, Japan; kenkana@cocoa.ocn.ne.jp; 4Department of Home Care Nursing, St. Luke’s International University, 10-1 Akashi-cho, Chuo-ku, Tokyo 1040044, Japan; masaymd@slcn.ac.jp; 5Department of Rehabilitation Medicine I, School of Medicine, Fujita Health University, 1-98 Dengakubo, Kutsukake-cho, Toyoake, Aichi 4701192, Japan; hkagaya2@fujita-hu.ac.jp (H.K.); esaitoh1@me.com (E.S.); 6Japanese Red Cross Toyota College of Nursing, 12-33 Nanamagari, Hakusan-cho, Toyota, Aichi 4718565, Japan; y-kamakura@rctoyota.ac.jp; 7Global Nursing Research Center, Graduate School of Medicine, The University of Tokyo, 7-3-1 Hongo, Bunkyo-ku, Tokyo 1130033, Japan; yohei-tky@umin.ac.jp; 8Department of Biostatistics, School of Public Health, Graduate School of Medicine, The University of Tokyo, 7-3-1 Hongo, Bunkyo-ku, Tokyo 1130033, Japan; matuyama@epistat.m.u-tokyo.ac.jp; 9Department of Gerontological Nursing/Wound Care Management, Graduate School of Medicine, The University of Tokyo, 7-3-1 Hongo, Bunkyo-ku, Tokyo 1130033, Japan

**Keywords:** aspiration pneumonia, community dwelling, residue, swallowing disorders, ultrasonography

## Abstract

The demand for methods to ensure safe oral consumption of food and liquids in order to prevent aspiration pneumonia has increased over the last decade. This study investigated the safety of swallowing care selected by adding ultrasound-based observation, evaluated its efficacy, and determined effective content of selected swallowing care. The study employed a pragmatic quasi-experimental research design. Participants were 12 community-dwelling adult patients (age: 44–91 years) who had experienced choking within 1 month prior to the study. After the control phase, in which conventional swallowing care was provided, trained nurses provided ultrasound observation-based swallowing care for a minimum period of 2 weeks. Outcome measurements were compared across three points, namely T1—beginning of the control phase, T2 and T3—before and end of the intervention phase. The mean durations of intervention were 30.8 days in the control phase and 36.5 days in the intervention phase. Pneumonia and suffocation did not occur in the control phase or the intervention phase. The safe intake food level and the food intake level score significantly improved during the intervention phase (*p* = 0.032 and 0.017, respectively) by adding eating training based on the ultrasound observation. However, there was no significant improvement in the strength of the muscle related to swallowing by the selected basic training. Our results suggest that swallowing care selected based on the ultrasound observation, especially eating training, safely improved safe oral intake among community-dwelling adults with swallowing dysfunction.

## 1. Introduction

Aspiration pneumonia is one of the leading causes of death in patients with swallowing dysfunction [1]. To avoid complications in individuals with a swallowing dysfunction, eating or drinking are greatly restricted to prevent aspiration pneumonia. This results in decreased muscle mass [2], malnutrition, and low quality of life [3,4]. However, as over 50% patients believe that relying on a feeding tube to live is the same as or even worse than death [5], a basic strategy in dysphagia care is to help patients with swallowing dysfunction continue to consume food and liquid safely for as long as possible [6]. This strategy also contributes to preventing community-dwelling patients from presenting into a hospital due to pneumonia caused by aspiration and pharyngeal residues.

Healthcare professionals, such as nurses or care givers, especially in the home care setting, should select swallowing care for dysphagia patients based on their swallowing function, and risk of aspiration and pharyngeal postswallow residue [6]. The gold standard methods for direct observation of aspiration and pharyngeal postswallow residue include the videofluoroscopic swallowing study (VFSS) and fiberoptic endoscopic evaluation of swallowing (FEES) [7,8]. Direct observation such as VFSS and VEES is usually provided by physicians or dentists in Japan [9,10]. The Japanese healthcare system creates a difficult situation for dysphagia care in the community setting. Speech language therapists are not allowed to perform these methods legally in Japan. Furthermore, most of them work at hospital. On the other hand, nurses are first-line providers of swallowing care, and have the potential to perform VFSS, but the education of VFSS has just started [11]. Thus, nurses have to ask physicians to perform these methods when the swallowing function is speculated to be worsening or improving. However, nurses often hesitate to ask physicians to perform VEES or VFSS, knowing the disadvantages of direct observation method. VEES has physical adverse effects by the use of ionizing radiation [12]. Also, patients often refuse to receive VFSS because of pain and discomfort due to the insertion of an endoscope in the patient’s nose. Therefore, nurses have some difficulties providing swallowing care because correct information is lacking about the patient’s swallowing function more than 1 month until physicians perform VESS or VFSS [13]. 

Ultrasonography has been used for the observation of swallowing function by hyoid bone displacement and the volume of muscles related with swallowing [14,15]. Since there has been no standard procedure to observe aspiration or pharyngeal residue by ultrasonography, we recently developed a ultrasound-based direct observation method for this purpose [16,17]. We also confirmed that ultrasound can observe the incidence of aspiration and presence or absence of pharyngeal residue with high sensitivity when VFSS was as references [16,17]. However, ultrasound observation is inferior to VESS or FEES as nonvisualization of some important information is observed by VESS or FESS (e.g., amount of residues, starting timing of swallowing). Although ultrasound observation is not a diagnostic measure, it is useful to screen out risk of aspiration and residue and to guide swallowing care for preventing these outcomes. Our previous study [13] demonstrated the efficacy of recommendations for swallowing care utilizing the guide by ultrasound observation. Nevertheless, this study has some shortcomings. Firstly, the researcher just gave healthcare professionals recommendations about continuation or change of food type or positioning, or advice to add FEES by the physician in charge. It is needed to evaluate which of swallowing care guided by ultrasound observation is effective to consume food and liquid safely. Secondly, this study was conducted by the researcher who was well-trained to use the ultrasound to monitor aspiration and pharyngeal residues. Accuracy and safety of ultrasound observation by general nurses well-trained to utilize were unknown. Therefore, this study aimed to confirm the safety of swallowing care selected by adding ultrasound-based observation, evaluate its efficacy, and determine effective content of selected swallowing care.

## 2. Materials and Methods

### 2.1. Study Design and Participants

The study was conducted in Hakodate City in Japan from February to August 2018 and used a pragmatic quasi-experimental research design. The study is registered with the University Hospital Medical Information Network Clinical Trials Registry, number UMIN000035140. After more than 1 week of administering conventional swallowing care, the new swallowing care regime comprising ultrasound-based observation of the residue in the epiglottis valley was administered for a minimum period of 2 weeks (Figure 1). 

Community-dwelling adults who met the following criteria were included—(1) had been receiving visiting nursing care, (2) had experienced choking within the last 1 month, (3) had been suspected of having dysphagia by visiting nurses, and (4) wished for oral intake. The exclusion criteria were the presence of a tracheotomy hole, systemic infection, or psychiatric disease. They were introduced into the research by home-visit nurses, a physician in charge of medical treatment, or managers in nursing facilities.

Each participant received a participant information sheet and oral explanations regarding the study, and provided written informed consent prior to participation in the study. If a participant had cognitive impairments or was unable to provide a signature on the consent sheet due to motor dysfunction, researchers received consent from a family member who was given complete information about the study protocol. The study protocol was approved by the Ethics Committee of the Graduate School of Medicine, The University of Tokyo (No. 11853-(2)). 

### 2.2. Swallowing Care Regime

We provided education programs with observational structured clinical examinations for two nurses. The two nurses monitored swallowing function in the recruited patients, and planned and provided swallowing care. Hand-held ultrasonography devices with 10 MHz linear transducers (SonoSite iViz, Fujifilm, Tokyo, Japan) were used for ultrasound observation.

During the control phase, the nurses administered conventional swallowing care (as shown in Table 1) based on patients’ subjective symptoms or observations, such as choking and wet hoarseness during eating or drinking. They observed the residue in the epiglottis valley using ultrasound just as an outcome measurement and did not use the evaluation of the ultrasound images for the determination of swallowing care.

At the beginning of the intervention phase, the swallowing care regime for all the patients was planned based on the algorism (Figure 2) which adds ultrasound observation into the conventional observation. The swallowing care regime could include basic and eating training [18]. Same content of basic training was recommended for all patients in the algorism. Adding eating training was recommended by the algorism when the following signs were observed—(1) residue in the epiglottis valley observed by ultrasound, or (2) choking and/or wet hoarseness. 

Basic training included head-lifting exercises and basic exercises comprising deep breathing, neck stretching, winding shoulder movements, puffing up one’s cheeks, in-and-out movement of the tongue, and repeated pronunciation of a specific word, namely “pataka,” that results in the closing of the nasal cavity. Basic exercises, such as neck stretching and winding shoulder movements, contributes to the relaxation of muscles related with swallowing function (e.g., suprahyoid muscles) and it also leads to arousal for individuals when they perform it before meals [18]. Patients were asked to record a daily training diary and to regularly perform head-lifting exercises and basic exercises every before mealtime (i.e., three times per day).

Eating training included 4 components—(1) diet modifications (choice of food texture appropriate to the swallowing function level), (2) pacing and feeding strategies (small volume of food per swallow, modifying the bolus size), (3) adjustment of posture during eating (increasing stability of the trunk using cushions and using a chin-down posture during swallowing [19]), and (4) clearance of residue by alternative swallowing or suctioning after the meal. We limited the oral intake level where residue was not detected by ultrasound when nurses planned the eating training considering safety.

After receiving permission to proceed with the swallowing care regime (i.e., both eating and basic training) from the physician in charge of the patient’s medical treatment, the nurses introduced the participants, primary caregivers, and healthcare staff in the facilities to the training process. When other health professionals had already been providing swallowing care, the nurses shared results of the ultrasound observation of residues in the epiglottis valley. Next, the nurses discussed the algorithm-based swallowing care plan with the other healthcare professionals before reporting it to the physician. The participants, family caregivers, and healthcare staff in the facilities executed the planned swallowing care and the nurses monitored the residue in the epiglottis valley by ultrasound at least once per two weeks during the intervention phase. When nurses confirmed that the eating based on the planned swallowing care was safe, and the patients demonstrated an ability to eat more complex food textures, the food texture level was gradually raised while continuously monitoring the residue using ultrasound.

### 2.3. Outcome Measurements

Both primary and secondary outcomes were collected by research assistant nurses. The primary outcome was the incidence of pneumonia or suffocation. Based on regular clinical judgment and practice [20], a physician blinded to the phase of the protocol diagnosed and treated any symptomatic pneumonia. The physician used blood tests and X-rays if pneumonia was suspected. We collected episodes of suctioning as an indicator of suffocation from the nursing record, which was shared among patients, family caregivers, and healthcare staff in the facilities.

The secondary outcomes included safe intake food level, daily intake food level, and strength of the muscle related to the swallowing function at three periods (T1: First observation of the control phase, T2: Observation immediately before the intervention phase, T3: Last observation of the intervention phase; Figure 1).

We originally developed an evaluation tool of safe intake food level. The safe intake food level was defined (a) as the most complex food level inducing minimum postswallowing residues as observed by ultrasound, or (b) as the food level at which the nurse stopped providing more complex food with more solid texture in order to avoid aspiration by residue in the epiglottis valley while eating test food or a daily meal. The safe intake food level was classified into 7 levels on the standard criteria of food texture in Japan [21] (ranging from 0—residue after swallowing the lowest level of food texture, i.e., jelly; to 6—no residue after swallowing any kind of food texture). The nurses rated the safe oral intake level based on observations of residue in the epiglottis valley by ultrasound, choking and wet hoarseness. 

The daily intake food level is a 10-level food intake level scale (FILS) [22]. The nurses collected pictures of the daily meal or the record in the food diary written by the patients or caregivers, and a researcher converted them into the 10-level FILS.

The strength of the muscles related to swallowing, such as suprahyoid muscles, was evaluated based on (a) the duration for which the patient kept his/her head up in the supine position, and (b) the number of times the patient repeated the word “pataka” in 10 seconds, a conventional oral diadochokinetic task [23].

Demographic data such as age, sex, place of residence were obtained from the medical charts or through direct interviews with the participants, family members, or care staff in the facilities.

### 2.4. Statistical Analysis

Descriptive and comparative analyses were performed using IBM SPSS version 23.0 (SPSS Inc., Chicago, IL, USA). Individual changes in safe oral intake and muscle strength were compared among three periods (T1, T2, and T3) using repeated-measures analysis of variance (ANOVA) followed by Tukey’s post hoc test for the correction of multiple comparisons. Results with two-sided p-values less than 0.05 were considered statistically significant.

## 3. Results

Of the 17 patients recruited, the 12 patients who followed the regime for more than 2 weeks in the intervention phase were included in the final analysis. The mean durations for which the regime was followed were 30.8 days (range, 7–56 days) in the control phase and 36.5 days (range, 14–56 days) in the intervention phase. Patients received ultrasound observation 0.78 ± 0.20/week during the intervention phase. The mean age of the patients was 71.5 years, and 7 patients (58.3%) were classified under the severe care needs category (Table 1). Four patients (33.3%) had been diagnosed with aspiration pneumonia within the 1 year preceding the study. 

Seven patients showed residues in the ultrasound images and four patients choked and/or experienced wet hoarseness during the control phase. The swallowing care regime was selected based on the algorithm; the nurses modified the swallowing care regime in patients who could not perform basic training, posture adjustment, or residue clearance due to cognitive or physical impairments (Table 2). All patients who were prescribed basic training performed the training three times in more than 40 days.

Pneumonia or suffocation did not occur in the control phase or the intervention phase. The safe intake food level significantly increased during the intervention phase (T2, 3.5 ± 2.1; T3, 5.1 ± 1.9; *p* = 0.032, Table 3) and FILS also significantly increased during the intervention phase (*p* = 0.017). Further, the daily intake food level improved with intervention in seven patients (58.3%). However, neither the duration for which patients could keep their head up (T2, 15.9 ± 9.1; T3, 726.4 ± 24.8; *p* = 0.222) nor the number of times they could repeat the word “pataka” (T2, 12.2 ± 4.1; T3, 13.6 ± 4.4; *p* = 0.055) were significantly improvemed among the five patients who performed basic training.

## 4. Discussion

This study revealed that the swallowing care regime involving ultrasound-based monitoring of residues in the epiglottis valley was safely provided without incidence of pneumonia among community-dwelling adults with decreased swallowing function. This study also showed the swallowing care by adding ultrasound-based observation improved oral intake levels. Through the study, eating training was determined as the effective content to improve the daily intake food level.

It is well known that inappropriate or ineffective swallowing care for patients is associated with severe risks such as suffocation. While swallowing care is usually requested in a home-visiting care setting, it is very challenging to conduct research to evaluate new swallowing care methods since home-care patients are in a relatively severe health condition. Despite its small sample size and lack of a control group, this study has significant meaning in the context of showing the necessity and possibility of the development of a safe and effective swallowing care regime by well-trained general nurses for such patients.

An important consideration while introducing a new assessment technique during swallowing care is avoiding aspiration pneumonia and suffocation related to incorrect assessment. In this study, pneumonia and suffocation did not occur in the intervention phase even though the daily intake food level was raised in over 50% of the participants. Furthermore, the change in other factors that are related to the incidence of aspiration pneumonia (i.e., deterioration of physical condition) were not observed among patients. Considering that aspiration pneumonia occurs within a week of aspiration [24], we set our minimum follow-up period to 2 weeks in the intervention phase. Our findings suggest that ultrasound observation of postswallowing residues in the epiglottis valley and algorithm-based swallowing care by well-trained general nurses would not increase the risk of aspiration pneumonia.

Although only five participants received basic training and the strength of the muscle related to swallowing did not improve after more than 40 days of training in the intervention phase, the daily intake food level was seen to significantly increase even in the two weeks of intervention (ID:16). Previous studies [25,26] have reported that including visual assessment of swallowing function using FEES in the swallowing care regime enabled raising of the food texture level and the commencing of oral intake in patients. Assessment technique consists of objective observation, suspected pathophysiology of swallowing dysfunction, and replanning care based on an algorism. The success of this study may be owing to the fact that nurses could accurately assess swallowing function through ultrasound-based direct observation of residues in the epiglottis valley. When decreased swallowing function is suspected by means of subjective symptoms such as choking or wet hoarseness, nurses and care staff are usually afraid of the risk of aspiration or suffocation [27] and consequently opt for conservative care. Ultrasound-based direct observation of residues in the epiglottis valley provides nurses with the confidence and security essential for them to raise the food level in the daily meal regardless of the frequency of intervention.

Removal of residues is an important aspect of swallowing care [28] because it is a leading cause of silent aspiration [29]. Residue is a sign of swallowing dysfunction related to decreased sensation in the pharynx and weakened swallowing pressure. Thus, food or liquid in the pyriform sinus and epiglottis valley may easily drop into the trachea without a swallowing reflex in patients with swallowing dysfunction. In this study, patients were able to continue eating safely because nurses were able to identify patients at the risk of silent aspiration through ultrasound observation and provide appropriate care to clear the residues after the meal. Notably, one of the patients (ID:4) originally on tube feeding in the control phase was able to start eating small amounts of pudding or jelly by suctioning the residue after the meal. Thus, accurate assessment of swallowing function through direct observation of residues would give patients the pleasure of eating food without the increased risk of aspiration pneumonia.

Eating training may be more effective than basic training to continue safe eating or drinking until the end of life because providing basic training is difficult in patients with severe cognitive impairment or physical dysfunction. Recently, various dysphagia diets developed through the use of new cooking methods have been made available in supermarkets and restaurants. Through appropriate postswallowing clearance care, correct choice of food texture, and the right seating posture during eating, adults with dysphagia can enjoy eating and their quality of life would significantly improve. Ultrasound observation has various advantages—(1) it is noninvasive in nature; (2) it provides a means for direct observation; and (3) it is a relatively simple method for use in a home care setting. Thus, these characteristics of ultrasound observation may contribute to enjoyment of eating and improve quality of life for community-dwelling adults with dysphagia.

Community settings have some specific features that healthcare professionals have to deal with. For example, the severity of dysphagia, primary cause of dysphagia, and the willingness to consume food or liquids orally are quite different among individuals. As people think restriction of oral intake is the same as or even worse than death [5], it is demanded to support the continuation of eating food and liquid safely for as long as possible at home. We set a priority on having willingness for oral intake among variation of community-dwelling patients in this study. Further, community-dwelling adults or older adults receive care services from several independent care professionals. Thus, sharing the patient’s health information, his/her will to eat and drink orally, and the primary goal of the swallowing care are of significant importance for interdisciplinary collaboration in these settings. Moreover, swallowing function may easily change in relation to daily physical conditions, such as the common cold. Thus, nurses and care staff must be capable of modifying the swallowing care regime in a timely manner. In the future, when ultrasound images are shared among health care professionals in the community setting through ICT, appropriate swallowing care can be smoothly administered and tested in patients with dysphagia.

This study has several limitations. First, we did not control the duration of the research period and the frequency of ultrasound observation. The severity level of swallowing dysfunction varied considerably across participants because we had to preinitialize the intention of patients, family and caregivers pragmatically. Unfortunately, the daily meal food level decreases with decreasing swallowing function in the long-term. Further studies are needed to evaluate the effectiveness of ultrasound-based swallowing care in maintaining the safe oral intake level with gradually decreasing swallowing function. Second, the nurses observed residues in the epiglottis valley during and just after eating meals. Backwash of food particles from the stomach after the meal is also a serious cause of aspiration pneumonia [29] that has not been addressed in this study. Observation of chronic secretions in the pyriform sinuses might contribute to increasing the effectiveness of our regime in the prevention of aspiration pneumonia. Third, we did not observe aspiration and residue in the pyriform sinus. The pyriform sinus one of the pharyngeal sites where aspiration and residues are observed, and it is a very close area to trachea. As the trachea contains air, which causes ultrasound attenuation, visualizing the pyriform sinus takes a long time for nurses who have just completed the education program. On the other hand, the epiglottis valley is easily visualized by regarding epiglottis as a typical landmark without attenuation. Sufficient sensitivity and specificity have been confirmed for the observation of the residue in the epiglottis valley [10]. Considering these advantages and disadvantages for both observation sites, it is feasible for nurses to repeatedly observe residues in the epiglottis valley during mealtimes. Fourth, participants of this study were not controlled by severity of dysphagia or primary cause of dysphagia. These factors are important for the effectiveness of swallowing care. Further study including a range of patients can be started since this study has shown the safety of swallowing care adding ultrasound observation. Finally, the success of our study might be related to increased motivation for and knowledge of swallowing care among nurses. The nurses who participated in the study received an education program related to ultrasound observation that enabled them to better understand the pathophysiology of dysphagia and means of swallowing care. 

## 5. Conclusions

This study confirmed that swallowing care based on an algorithm involving ultrasound observation of residues in the epiglottis valley by nurses can be safely implemented. This study also showed the efficacy of swallowing care by adding ultrasound-based observation for improving oral intake levels in community-dwelling adults with swallowing dysfunction. In addition, the study revealed that eating training based on the ultrasound observation of swallowing function may be more effective than basic training to improve the daily intake food level in these patients.

## Figures and Tables

**Figure 1 healthcare-08-00050-f001:**
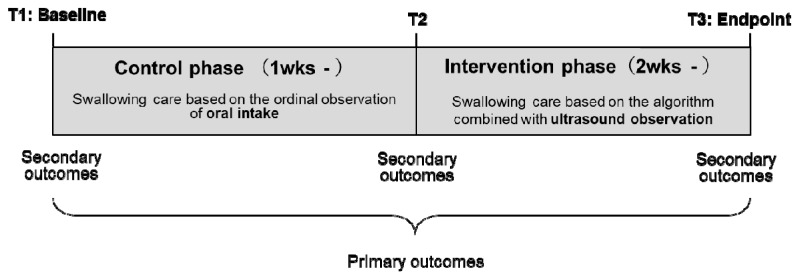
Study design.

**Figure 2 healthcare-08-00050-f002:**
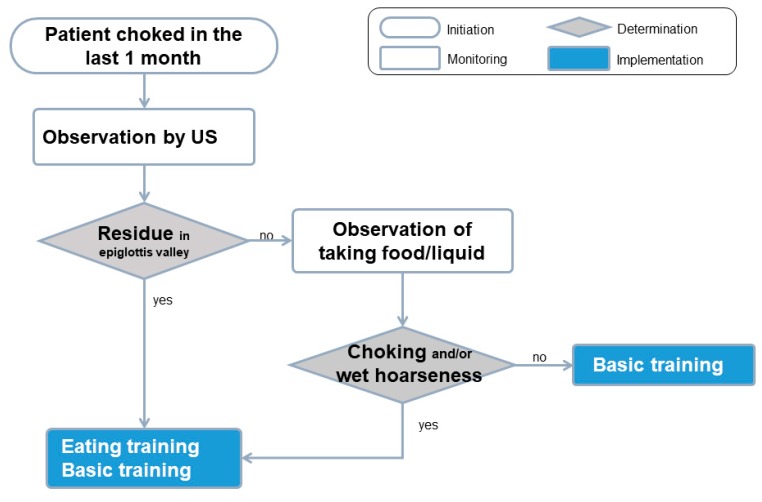
The swallowing care algorithm including ultrasound observation.

**Table 1 healthcare-08-00050-t001:** Demographic characteristics of patients (*n* = 12).

Characteristics	Mean (range) or *n* (%)	
Age (years)	71.5 (44–91)	
Sex (number of males)	9	(75.0%)
Living place		
Home	6	(50.0%)
Facility (e.g., nursing home)	6	(50.0%)
Certified care level		
Level 4 or 5 (bedridden)	7	(58.3%)
Less than Level 3	4	(33.3%)
Not certified	1	(8.3%)
Insurance for home visiting nursing		
Long-term care insurance	11	(91.6%)
Health care insurance	1	(8.3%)
Primary medical condition or impairment		
Dementia	5	(41.6%)
Neurogenic/muscle disease	2	(16.6%)
Stroke	1	(8.3%)
Terminal stage	1	(8.3%)
Cervical spinal injury	1	(8.3%)
Cerebral palsy	1	(8.3%)
Frailty	1	(8.3%)
History of aspiration pneumonia within last 1 year (yes)	4	(33.3%)
Intake status		
Oral intake only	9	(75.0%)
Tube feeding mainly	1	(8.3%)
Tube feeding only	2	(16.6%)
Oral intake status (*n* = 10)		
Regular	5	(50.0%)
Chopped	2	(20.0%)
Soft	0	(0%)
Mousse	3	(30.0%)
Swallowing care (multiple answers)		
Basic training	2	(16.6%)
Diet modification	5	(41.6%)
Pacing and feeding strategies	0	(0%)
Adjustment of posture during eating	2	(16.6%)
Clearance of residue after meal	1	(8.3%)

**Table 2 healthcare-08-00050-t002:** Swallowing care.

Swallowing care	Algorithm ^§^	Cognitive or Physical Impairments	
		No (*n* = 5)	Yes (*n* = 7)
**a. At least one of (a) presence of residue, (b) choking, or (c) wet hoarseness**		***n* = 5**	***n* = 6**
Eating training			
Diet modification	Recommended	1 (20%)	6 (100%)
Pacing and feeding strategies	Recommended	5 (100%)	6 (100%)
Adjustment of posture during eating	Recommended	4 (80%)	3 (50%)
Clearance of residue after meal	Recommended	5 (100%)	3 (50%)
Basic training			
Head-lifting exercises	Recommended	5 (100%)	Impossible
Basic exercises	Recommended	4 (80%)	Impossible
**b. Absence of residue, choking, or wet hoarseness**		***n* = 0**	***n* = 1**
Eating training			
Diet modification	Not recommended		0
Pacing and feeding strategies	Not recommended		1 (100%)
Adjustment of posture during eating	Not recommended		1 (100%)
Clearance of residue after meal	Not recommended		0
Basic training			
Head-lifting exercises	Recommended		Impossible
Basic exercises	Recommended		Impossible

^§^ Algorithm: Recommended swallowing care based on the ultrasound-based algorithm.

**Table 3 healthcare-08-00050-t003:** Safe oral intake scores during the experiment (*n* = 12).

ID	Intervention Period (days)	Safe Intake Food Level			FILS ^§^		
		T1	T2	T3	T1	T2	T3
ID3	47	1	1	6	3	3	4
ID4	28	0	1	1	1	1	3
ID6	42	2	4	6	8	8	9
ID7	46	4	4	6	8	8	9
ID9	56	1	1	6	8	8	9
ID11	50	6	5	6	8	8	9
ID13	51	0	4	6	8	8	8
ID14	31	1	1	1	7	7	7
ID15	45	6	6	6	9	9	9
ID16	14	6	6	6	4	4	9
ID17	14	2	6	6	8	8	8
ID18	14	5	3	5	7	7	7
Total (mean ± SD)		2.8 ± 2.4	3.5 ± 2.1	5.1 ± 1.9 ^⧺,^^⧻^	6.6 ± 2.5	6.6 ± 2.5	7.6 ± 2.1 ^⧺^

^§^ FILS, food intake level score. T1, first observation of the control phase; T2, observation just before the intervention phase; T3, last observation of the intervention phase; SD, standard deviation; ID, patient identifier; 0, residue after swallowing the code 0j (jelly form for swallowing training with low protein content, homogeneous physical properties, low adhesion, high cohesiveness, soft hardness, and minimized syneresis (e.g., soft jelly)); 1, residue after swallowing the code 1j (jelly form of texture modified diet with homogeneous physical properties, low adhesion, high cohesiveness, soft hardness and minimized syneresis (e.g., pudding, mousse)); 2, residue after swallowing the code 2-1 (puree form with smooth, homogeneous physical properties (e.g., vegetable puree)); 3, residue after swallowing the code 2-2 (puree form with smooth, heterogeneous physical properties (e.g., meat puree)); 4, residue only after swallowing code 3 (moist form with easy bolus formation, minimized syneresis and considerable cohesiveness); 5, residue only after swallowing code 4 (minced form with easy bolus formation and hard to spread); 6, no residue after swallowing any food texture. ^⧺,^^⧻^ Repeated-measure analysis of variance with Tukey’s post hoc test: ^⧺^ T2 vs. T3: *p* < 0.05, ^⧻^ T1 vs. T3: *p* < 0.05.

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
