# Peer review of "Effectiveness of Swallowing Care on Safe Oral Intake Using Ultrasound-Based Observation of Residues in the Epiglottis Valley: A Pragmatic, Quasi-Experimental Study"

_healthcare, 2020, doi:10.3390/healthcare8010050_

Round 1

Reviewer 1 Report

I like the topic of this paper and I do think that it has value. 

The following areas were challenges for me: 

The writing style became a bit verbose which made it difficult to follow at times.  The main focus of this research according to the title was the use of ultrasound based observations when detecting residue, however, when reading the study, I feel that this was not actually the focus. The methodology and the results spend more time focusing on the intervention strategies used rather than the use of ultrasound. Due to this, I am not sure how the researchers can make the conclusions about the use of ultrasound based observations. Overall, there were too many components and was not too sure what the argument was and the conclusions were not made clear.  The methodology section lacked significant detail especially when it came to the outcome measures and tools used.  

Author Response

Thank you very much for your generous comments. According to your suggestions, we extensively corrected our manuscript as follows. All the corrected parts were highlighted by red color characters.

Reviewer 2 Report

Thank you for submitting this interesting article. The content has the potential to be of interest to the readership however some revisions would strengthen the paper.

1) The paper needs to be checked for grammar eg line 68 obtained, line 90 need to add suspected 'of'

2) Line 75 add text to explain why it is easier to see residue in the epiglottis valley than the pyriform sinuses. Expand detail around sensitivity and specificity of using ultrasound to see residue. What scoring system did you use to rate the degree of residue?

3)Line 137 give an explanation and evidence for relevance of neck stretching and winding shoulder movements

4) Line 150 were nurses identifying aspiration from ultrasound? What is reliability of this? If not how was aspiration identified?

5) Line 157 Which swallowing muscles are you referring to? Which muscles are you specifically working on and what is the evidence to support this?

6) Line 172 explain why the observation period ranged so widely from 7-56 days? If this is pre-intervention why is the period so wide. How often per day were people observed and for how long?

7) Explain the wide variation in the intervention phase. You do not appear to be comparing like with like. How often per day was the intervention and for how long? Could intensity of intervention had an effect? How may repetitions of the exercises were people doing? State the evidence to support your dosage.

8)The discussion over states your findings, please amend. Aspiration pneumonia is multi-faceted you cant say assessment caused no aspiration pneumonia. Effects of the frequency of the intervention needs explaining in more detail in your discussion.

9) The conclusions are more accurate but will need to reflect the additions to your discussion section.

Author Response

(The authors gave the same response as above.)

Round 2

Reviewer 1 Report

I have tried to make my comments as helpful as possible. I explained my understanding of what is written and asked questions that perhaps show the gaps in in my understanding. There were a lot of details that required clarity. It does require extensive re-working. 

Reviewer 2 Report

Thank you for your revisions. At present they are somewhat confusing for the reader and should be re-phrased more clearly in the English language. 

I would remove the section of the limitations of the study from the introduction and discuss these in more detail in the discussion section.
